# Comprehensive Analysis of the Association between Human Diseases and Water Pollutants

**DOI:** 10.3390/ijerph192416475

**Published:** 2022-12-08

**Authors:** Xinlu Jiang, Huanhuan Zhang, Xiaoyan Wang, Xu Zhang, Kaiyang Ding

**Affiliations:** 1Department of Hematology, Anhui Provincial Hospital Affiliated to Anhui Medical University, Hefei 230032, China; 2Department of Respiratory, Wannan Medical College, Wuhu 241002, China; 3Department of Urology, The First Affiliated Hospital of Nanjing Medical University, Nanjing 210029, China

**Keywords:** water pollutants, diseases, phenotypes and pathways, toxicogenomic analysis

## Abstract

Drinking water is an important natural resource. For many people worldwide, especially in developing countries, access to safe drinking water is still a dream. An increasing number of human activities and industrialization have caused various physical, chemical, and biological pollutants to enter water bodies, affecting human health. Water pollutants contain a vast number of additives, such as perfluorinated chemicals, polybrominated diphenyl ethers, phthalate, nanomaterials, insecticides, microcystins, heavy metals, and pharmacologies. In this work, we aim to explore the potential relationship between water pollutants and human diseases. Here, we explored an integrative approach to identify genes, biological processes, molecular functions, and diseases linked to exposure to these water pollutants. These processes and functions affected by water pollutants are related to many diseases, including colonic neoplasms, breast neoplasms, hepatitis B, bladder cancer, and human cytomegalovirus infection. In addition, further analysis revealed the genes that play a key role in the human diseases induced by water pollutants. Therefore, conducting an integrative toxicogenomic analysis of water pollutants is more appropriate for evaluating the potential effects of water pollutants on human health.

## 1. Introduction

Water is one of the most important resources to mankind. With the increasing level of industrialization and urbanization, it becomes more and more important due to the increasing demand for industrial and domestic water [1]. However, the availability of global water resources varies greatly in different countries and regions. What is worse, climate change and water pollution further aggravate the crisis of water shortage [2]. As reported by the World Health Organization, approximately 2 million people in the world do not have access to clean water [3]. Moreover, it is estimated that about 3.4 million people die each year due to water shortages or water contamination, and 99% of these deaths occur in developing countries [4]. Since water is the most active natural element involved in the transformation of ecosystems, water quality greatly affects human health. However, due to the pollution of chemicals and solid waste, water often becomes the carrier of infectious and communicable diseases [5].

Over the past decades, water contamination has become a worldwide problem, which is increasingly threatening our health [6]. Despite making significant progress in the management and treatment of water in recent years, levels of pollutants varied widely, and the methods of disinfection and filtration were different [7]. Drinking water continues to often contain a lot of harmful substances. In 2010, the oil spill in the U.S. Gulf of Mexico aroused great attention from the international community. Although the U.S. government has invested a lot of manpower and material resources to deal with pollution as soon as possible, the pollution caused the destruction of 1000 miles of wetlands and beaches along the Gulf of Mexico [8]. Recently, the Japanese government announced plans to discharge diluted wastewater into the sea after the Fukushima nuclear wastewater tank reached its peak storage capacity, triggering fierce criticism from the outside world and local fishermen. Severe water pollution incidents, such as the two mentioned above, have undoubtedly led to a sharp decline in biodiversity, and the toxins produced by the death of these marine organisms affected humans through the food chain [9]. In the 20th Pollutant Responses in Marine Organisms, a large range of chemicals, such as pesticides, pharmaceuticals and personal care products, endocrine disruptors, polybrominated diphenyl ethers, perfluorinated chemicals, and dibutyl phthalate, were considered important contaminants [10]. These pollutants appeared a long time ago but have recently been considered contaminants that seriously threaten the water environment. Because of the universality and adverse effects of these contaminants, these pollutants tend to be easily absorbed in rivers, lakes, and oceans. In recent studies, these water pollutants have been proven to be closely associated with many diseases [11]. Long-term exposure to persistent organic pollutants increases the risk of cardiovascular disease. Moreover, the prolonged exposure to heavy metals result in the death of neurons and finally lead to neurodegenerative disorders, such as Alzheimer’s disease, Parkinson’s disease, and Prion disease [12]. These water pollutants are ubiquitous and accumulate through the continuous aquatic food chain, causing adverse effects on human health [13].

The Comparative Toxicogenomics Database (CTD), a literature-based and manually curated public resource, aims to advance understanding of how environmental exposures affect human health. It provides manually curated information about chemical–gene/protein interactions, chemical–disease and gene–disease relationships. In this work, we aimed to evaluate the gene interactions of water pollutants and explore the possible mechanism between water pollutants and diseases. Gene Ontology (GO) and Kyoto Encyclopedia of Genes and Genomes (KEGG) functional annotation analysis was applied to evaluate their effects on diseases, biological process, and expression in human tissues. We hoped to find more associations between diseases and water pollutants. Moreover, when dealing with health issues, water management measures can demonstrate ways to lessen the impact of many infectious diseases and other health problems.

## 2. Methods and Materials

### 2.1. Datasets Download

The gene, protein, and disease interactions data were obtained from the CTD database (http://ctdbase.org/ (accessed on 14 March 2021))) in March 2021. We collected almost all relevant information about water pollutants. Specifically, 51 chemical compounds were involved in 8 groups according to their industrial classifications. Next, we used the R to calculate the interactive counts involved in interactive genes of the same type of water pollutants. For the CTD database, the gene interaction includes the increase and decrease of the expression level of mRNA. Genes with more than a total of 100 interactions were considered high-interaction genes.

### 2.2. Gene Ontology (GO) and Kyoto Encyclopedia of Genes and Genomes (KEGG) Pathway Enrichment Analysis

After searching the interaction genes of water pollutants in the CTD database, we selected the genes with more than 100 interactions as the water-pollutant-related genes. To explore the biological and molecular functions of the water-pollutant-related genes, “clusterProfiler”, “enrichplot”, “org.Hs.eg.db”, and “ggplot2” packages were used in R to perform GO and KEGG functional annotation pathway enrichment analysis. Biological process (BP), molecular function (MF), and cellular component (CC) in GO analysis were performed; *p* < 0.05 was selected as the threshold in the analysis. We also used “tidyverse”, “ggraph” and “tidygraph” packages in R to calculate a circle diagram.

### 2.3. Protein–Protein Interaction (PPI) Network

In this work, a protein–protein interaction network of pollutant-related genes was conducted by STRING (https://www.string-db.org/ (accessed on 16 March 2021)). The PPI network is composed of individual proteins through their interactions with each other. By analyzing the interaction of a large number of proteins in biological systems, it is possible to understand all aspects of life processes, such as biological signal transmission, gene expression regulation, energy and material metabolism, and cell cycle regulation. All the information about protein–protein interaction was downloaded. We chose a very high interaction score (interaction score ≥ 0.9) as the screening criterion. Then, Cytoscape 3.8.2 (University of California, San Diego, CA, USA) was performed to analyze and visualize the protein–protein interactive counts that are obtained from the STRING. Hub genes, which are defined as genes highly interconnected with nodes in a module, have been considered functionally significant. In this experiment, the genes with more than 20 degrees were considered hub genes.

### 2.4. Specific Analysis of Water Pollutants According to Industry Classification

In the previous analysis, we performed the overall analysis of the interaction genes of water pollutants. In order to further determine the pollutant-related genes affected by the different chemical compounds, Venn diagrams were used to search the co-interaction genes of all 8 pollutant groups.

### 2.5. Identification of Disease-Related Genes Caused by Water Pollutants

In order to better understand the main health consequences caused by different types of water pollutants, the most associated diseases were obtained from co-interaction genes which were selected from each set of water pollutants. The gene–disease interactions data were obtained from the CTD database. The “clusterProfiler”, “tidyverse”, “data.table”, “ggraph”, and “tidygraph” packages were used in R to calculate a circle diagram. 

### 2.6. The Distribution of Hub Genes in Human Organs

In order to explore the differential expression of disease-related genes in human organs, human gene expression diagrams were used to evaluate the level of hub genes from the Human Protein Atlas website (https://www.proteinatlas.org/ (accessed on 17 March 2021)).

## 3. Results

### 3.1. Identification of Water-Pollutant-Related Genes

After fully searching the types of water pollutants from the CTD database, we obtained a total of eight types of water pollutants, including perfluorinated chemicals, polybrominated diphenyl ethers, phthalate, nanomaterials, insecticides, microcystins, heavy metal, and pharmacology. Among them, we selected 51 poisons that were closely related to water pollution (pollutants are shown in Figure 1). After the data processing, we obtained the total interaction counts about the water pollutants and interaction genes. Interaction genes with more than 100 interaction counts were considered to be highly related to water pollutants. In order to clarify the biological processes most affected by water pollutants, the genes most commonly affected by all water pollutants (based on the interaction counts collected by the CTD) were selected for further analysis. Among the interacting genes which were involved in water pollutants, a total of 77 genes showed more than 100 interaction counts. Among all included genes for further analysis, peroxisome proliferator-activated receptor alpha, PPARA (with 2102 interaction counts); tumor necrosis factor, TNF (with 779 interaction counts); interleukin-1 beta, IL1B (with 553 interaction counts); caspase-3, CASP3 (with 490 interaction counts); interleukin-6, IL6 (with 489 interaction counts); catalase, CAT (with 486 interaction counts); heme oxygenase 1, HMOX1 (with 406 interaction counts); copper-transporting ATPase 1, ATP7A (with 385 interaction counts); and prostaglandin G/H synthase 2, PTGS2 (with 333 interaction counts) showed the highest levels of interaction. 

### 3.2. GO and KEGG Pathway Enrichment Analysis

A total of 77 of the most interactive genes were uploaded in R to perform GO and KEGG functional annotation pathway enrichment analysis. and the annotation was selected as homo sapiens (Figure 2). The results of KEGG pathway enrichment analysis showed that the most common terms were related to lipid and atherosclerosis. In addition, the most significant diseases for these 77 interactive genes were kaposi sarcoma-associated herpes virus infection, Chagas disease, colorectal cancer, hepatitis B, bladder cancer, and human cytomegalovirus infection. Based on KEGG pathway enrichment analysis, among all the analyses, the tumors with the highest correlation with water pollutants were colorectal cancer, bladder cancer, pancreatic cancer, prostate cancer, and endometrial cancer. Phthalates have been shown to be a key factor in promoting the occurrence of colorectal cancer. Several studies have found that DBP can cause testicular damage and lead to a decrease in the number and quality of sperm. DBP, as a common phthalate, has been proven toxic to the male reproductive system. Exposure to DBP during adolescence can cause damage to the testicles, which in turn causes a decrease in sperm count and quality.

The GO functional annotation pathway enrichment analysis of interactive genes revealed that the BPs were most related to responses to metal ions, oxidative stress, chemical stress, drugs, and cadmium ions. Metal ion dyshomeostasis is a major characteristic of Alzheimer’s disease. One study suggested that the levels of serum metal ion are closely related to Alzheimer’s disease, and the clinical severity of Alzheimer’s disease patients is related to the level of serum metal ion concentration [14]. Oxidative stress can induce both apoptosis and cellular senescence. The damage of oxidative stress to macromolecules is significant because proteins and lipids are essential to maintaining the integrity of DNA/RNA and determining health and disease states. Much evidence supports the critical role of oxidative damage to macromolecules in the development of a variety of cancers [15].

For CC, the most common terms included membrane raft, membrane microdomain, membrane region, vesicle lumen, secretory granule lumen, transcription regulator complex, cytoplasmic vesicle lumen, platelet alpha granule lumen, and nuclear envelope. Membrane rafts are heterogeneous and dynamic domains characterized by a close packing of lipids [16]. The cell membranes of some solid tumors, such as breast and prostate cancer, contain higher levels of cholesterol, which means that larger rafts can be formed in these cell membranes. This may stimulate signaling pathways to promote tumor growth and progression [17]. Extracellular vesicles are one of the heterogeneous bilayer membrane vesicles released by all human cell types. In clinical use, extracellular vesicles have become a potential source of biomarkers for urological cancers [18]. 

Heme binding was the most common MF term. Tetrapyrrole binding, steroid hydroxylase activity, oxidoreductase activity, monooxygenase activity, DNA-binding transcription factor binding, and antioxidant activity were also considered to be significant MF terms. 

### 3.3. Construction of the PPI Network and the Identification of Key Genes

Next, in order to find key genes for the further exploration of the roles of interactive genes in water pollutants, cytoscape was used to analyze and construct a PPI network (Figure 3). The PPI network included 106 nodes and 992 edges. The degree value is correlated with node size, and the co-expression value is related to a small edge size. Genes with more than 20 degrees were screened as the biological hub genes. The PPI network of interactive genes showed that JUN, AKT1, TP53, IL6, RELA, MAPK1, MAPK3, FOS, TNF, and VEGFA may play an important role in diseases induced by water pollutants. Interestingly, all of these interaction genes and the proteins they encode show strong interconnections, suggesting that a change in transcription/function in one of them may affect others. All these hub genes show a certain correlation with other genes in the network, revealing that these genes may exert an impact on water-pollutant-induced diseases. 

### 3.4. Specific Analysis of the Different Type of Water Pollutants

In addition to the comprehensive analysis of the top 77 genes most affected by water pollutants (Figure 4). We next carried out the specific analysis according to their industrial classification. A total of eight of the most common water pollutants were classified, including perfluorinated chemicals, polybrominated diphenyl ethers, phthalate, nanomaterials, insecticides, microcystins, heavy metal, and pharmacologies. On the basis of their interaction counts of interaction genes, Venn diagrams were compiled to analyze the co-interaction genes in different types of water pollutants. Even though we classified the water pollutants according to the chemical species, due to the diversity of constituent chemicals, the number of co-interaction genes in the Venn diagram is not expected to be very high. Among perfluorinated chemicals and polybrominated diphenyl ethers, a total of 15 co-interaction genes and 14 co-interaction genes were identified, respectively. We obtained a total of 166 co-interaction genes in phthalate. Among heavy metals, a total of 56 co-interaction genes were found. Other water pollutants, such as nanomaterials, insecticides, microcystins, and pharmacologies, share very few co-interaction genes. Some water pollutants, such as phthalate and heavy metals, affect many co-interaction genes. Other water pollutants share few genes in common, which reveals that these compounds are not equally toxic and do not affect a multitude of genes in common.

### 3.5. Integrative Analysis of Disease-Related Genes

On the basis of PPI network results, interaction genes with the top 10 degrees were explored for further analysis. The PPI network demonstrated that JUN, AKT1, TP53, IL6, RELA, MAPK1, MAPK3, FOS, TNF, and VEGFA showed the most degrees (Figure 5). In order to further determine the relation between interaction genes and diseases, we then downloaded the gene–disease interactions data from the CTD database. Based on inference score, we focused on the 10 most related diseases of all the top 10 interaction genes. The correspondence between interaction genes and diseases is shown in the circle plot. The diseases which were strongly related to water pollutants were mainly hypertension, inflammation, neoplasm metastasis, and neoplasm invasiveness. Our results may indicate that the effects of the water pollutants are predominantly focused on very specific disease pathways. Among all tumor-related diseases, we discovered that colonic neoplasms and breast neoplasms were the most related tumors with water pollution, which is consistent with the results of the PPI network.

### 3.6. Identification of the Genes Related to Colonic Neoplasms and Breast Neoplasms

After analyzing the genes which were related to water pollutants, colonic neoplasms and breast neoplasms demonstrated close interaction. In order to specifically determine which genes were most related to colonic neoplasms and breast neoplasms, we downloaded the genes which highly interacted with colonic neoplasms and breast neoplasms from the CTD. On the basis of the inference score, a total of 32,452 genes were proved to relate to colonic neoplasms. We found that TP53 (ranked 5), RELA (ranked 8), and JUN (ranked 12) were the most interactive genes with colonic neoplasms. Based on the inference score, a total of 113,267 genes were proved to relate to breast neoplasms. TP53 and RELA were found to be the most relative to breast neoplasms. The distribution of TP53, RELA, and JUN in human organs are demonstrated in Figure 6. 

## 4. Discussion

Drinking water, as one of the very precious and important resources, is often not sufficient and easily available. Fresh water accounts for only 3% of the world’s water resources. However, this very small part of the world’s water resources meets the needs all the 1.8 million species [19]. Fresh water and freshwater biodiversity constitute valuable natural resources in economics, culture, aesthetics, science, and education. Therefore, their protection and management are of vital importance to the interests of all mankind, the country, and the government [20]. However, this precious legacy is under threat. The decline of freshwater biodiversity far exceeds the severely affected terrestrial ecosystems. If the demand for water resources remains high, species will continue to decrease at the current rate [21]. By using the gene-water pollutants data from the CTD database, our article discussed the pollutants in freshwater habitats and their harm to human health. With the help of in silico analysis, we conducted an integrated analysis of water pollutants, and we provided a bioinformatics approach that links water pollutants and disease. GO and KEGG enrichment analysis revealed that water pollutants are highly interactive with human diseases. On the basis of the CTD databases, we further explored the relationship between water pollutants and human tumors. Among all tumors, our analysis demonstrated that colonic neoplasms and breast neoplasms exhibited high interactions with water pollutants. Further investigations suggested that several interactive genes were strongly correlated with colonic neoplasms and breast neoplasms. Moreover, our research found that TP53, RELA, and JUN are strongly related to colonic neoplasms. In addition, TP53 and RELA might play an important role in breast neoplasms. 

This study was based on integrated analysis and specific analysis of water pollutants. In the integrated analysis of water pollutants, the top 77 interaction genes which are most frequently affected by water pollutants were selected. Among these interaction genes, PPARA was proven to be the gene with the most interaction. PPARA has been found to be closely involved in the process of DNA double-strand breaks repair, and this function may be related to the phenotype of apoptosis, cell death, and proliferation caused by exposure to water pollutants. TNF, another key gene with the second most interaction, has been identified as a key regulator of the inflammatory response and may increase leukocyte adhesion, endothelial migration, vascular leakage, and thrombosis. Therefore, agents that block the action of TNF have the ability to treat a range of inflammatory diseases, including rheumatoid arthritis, ankylosing spondylitis, inflammatory bowel disease, and psoriasis. Based on the enrichment analyses for these top 77 genes, we explored some interesting pathways that are related to vital biological processes, molecular function, cellular components, and diseases. When considering all water pollutants for analyzing the top 77 interaction genes in the KEGG pathway, atherosclerosis was shown to be the most related diseases. A rodent study found that acute exposure to malathion can cause disruption of lipid metabolism, increase LDL and triglycerides, and finally may increase the risk of atherosclerosis and cardiovascular disease [22]. Moreover, the urine di-(2-ethylhexyl) phthalate metabolite levels have also been found to be strongly associated with atherosclerosis [23]. In addition to atherosclerosis, the KEGG enrichment analysis demonstrated that numerous types of cancer were related to water pollutants, including colorectal cancer, breast cancer, bladder cancer, thyroid cancer, and pancreatic cancer. Moreover, it is interesting to find that water pollutants may also have a strong correlation with a large number of infectious diseases, such as human cytomegalovirus infection, hepatitis B, hepatitis C, toxoplasmosis, and yersinia infection. Next, a protein–protein interaction network was constructed to determine whether the interaction genes showed strong interconnections. The protein–protein interaction network analysis is important because it emphasizes the demand for a comprehensive analysis of water pollutants. On the basis of different chemical compounds according to that classification, we analyzed their co-interaction genes. On the one hand, some water pollutants shared lots of co-interaction genes, which may indicate that perfluorinated chemicals, polybrominated diphenyl ethers, and phthalate may demonstrate similar toxicity to human health. On the other hand, some water pollutants, such as nanomaterials, insecticides, microcystins, and pharmacologies show little co-interaction genes. This reveals that they are not similar in mechanisms of action in toxicity. In recent years, many emerging technologies have been applied in the detection of pollutants, which provided good directions for the reduction of water pollutants. A previous study utilized a microfluidic chip along with a real-time imaging system as an enabling technology for developing a portable system for on-site screening of multiple pathogens related to food and water safety [23]. In the future, combined bioinformatics analysis and new technologies can significantly reduce water pollutants.

## 5. Conclusions

According to the water pollution-related diseases that we have collected, these diseases are more common in the Western world than in the Eastern world. Water pollutants are disruptors of the endocrine system. Moreover, these pollutants are rare pollutants, and they are chemical compounds that humans are exposed to every day. Here, we have checked the risk of chronic exposure and conducted integrated and specific analysis of water pollutants.

## Figures and Tables

**Figure 1 ijerph-19-16475-f001:**
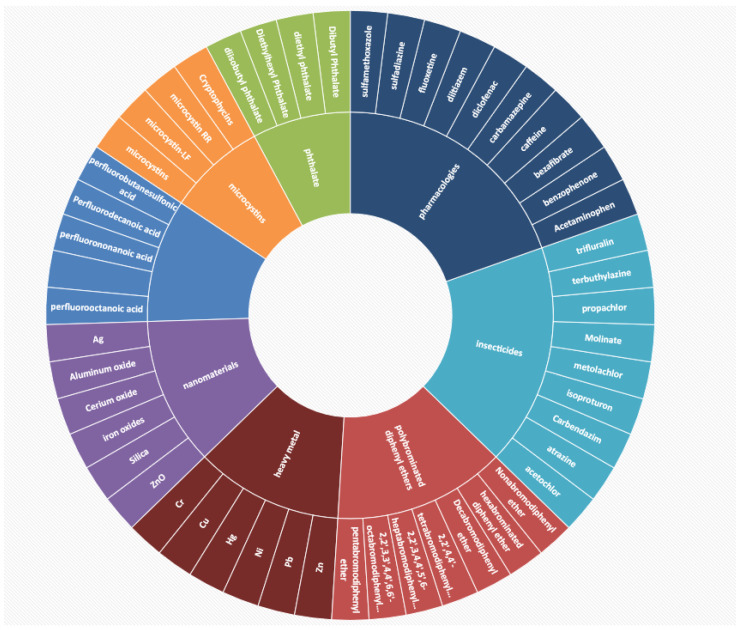
The most representative water pollutants analyzed in this work.

**Figure 2 ijerph-19-16475-f002:**
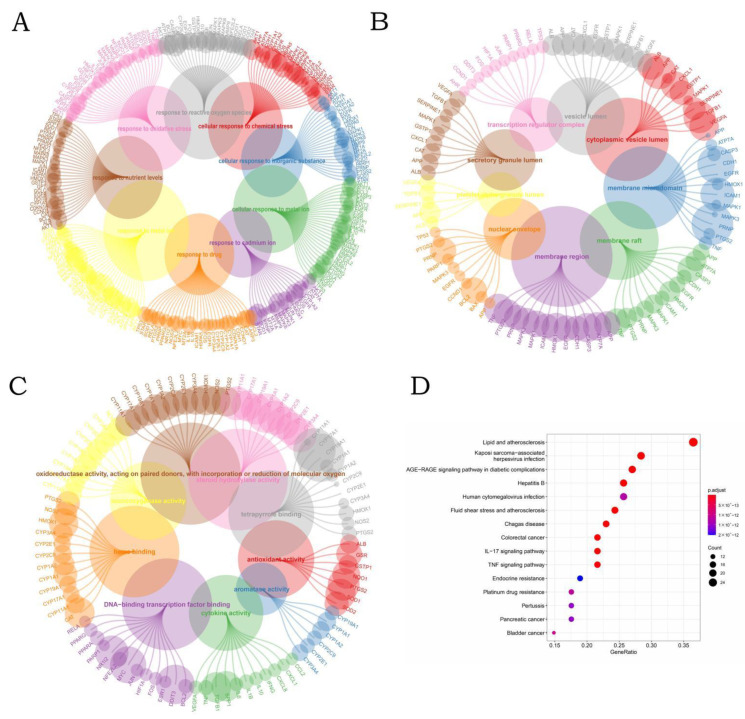
Circle plots were used to explore GO and KEGG enrichment pathways among the co-interaction genes: (**A**) top terms from the GO BP enrichment analysis; (**B**) top terms from the GO CC enrichment analysis; (**C**) top terms from the GO MF enrichment analysis; (**D**) top 4 terms from the KEGG enrichment analysis.

**Figure 3 ijerph-19-16475-f003:**
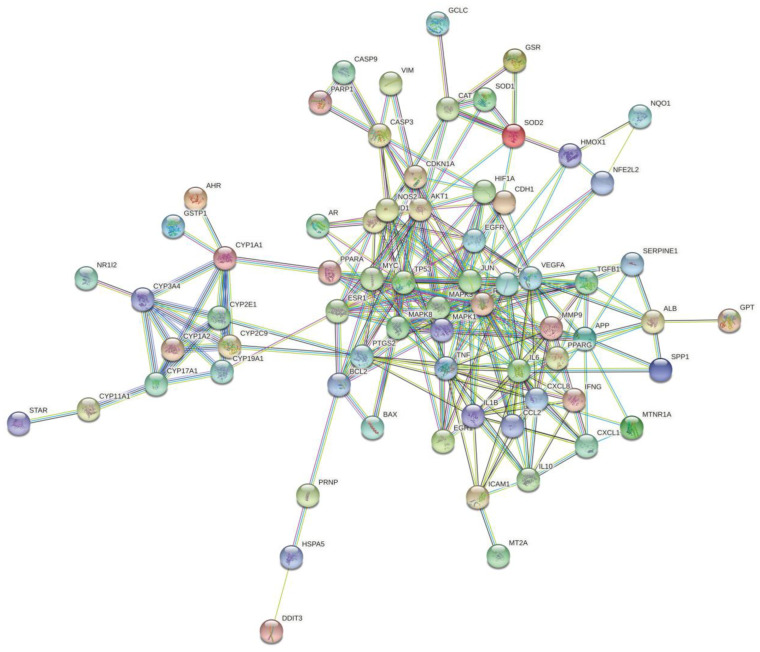
Construction of the PPI network and hub genes identified.

**Figure 4 ijerph-19-16475-f004:**
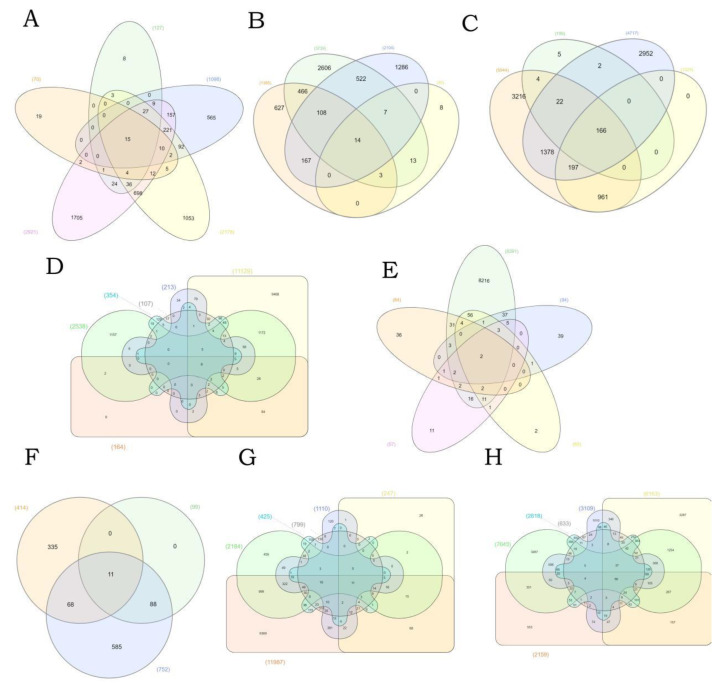
Venn diagrams of different types of water pollutants. The groups were created as follows: (**A**) perfluorinated chemicals: perfluorobutanesulfonic acid, perfluorodecanoic acid, perfluorooctane sulfonic acid, perfluorooctanoic acid, perfluorononanoic acid; (**B**) polybrominated diphenyl ethers: pentabromodiphenyl ether, 2,2′,4,4′-tetrabromodiphenyl ether, decabromodiphenyl ether, hexabrominated diphenyl ether; (**C**) phthalate: dibutyl phthalate, diethyl phthalate, diethylhexyl phthalate, diisobutyl phthalate; (**D**) nanomaterials: Ag, aluminum oxide, cerium oxide, iron oxide, silica, ZnO; (**E**) insecticides: acetochlor, atrazine, carbendazim, isoproturon, metolachlor; (**F**) microcystins: cryptophycins, microcystin RR, microcystin-LF, microcystins; (**G**) pharmacologies: acetaminophen, bezafibrate, caffeine, carbamazepine, diclofenac, fluoxetine; (**H**) heavy metal: Cr, Cu, Hg, Ni, Pb, Zn.

**Figure 5 ijerph-19-16475-f005:**
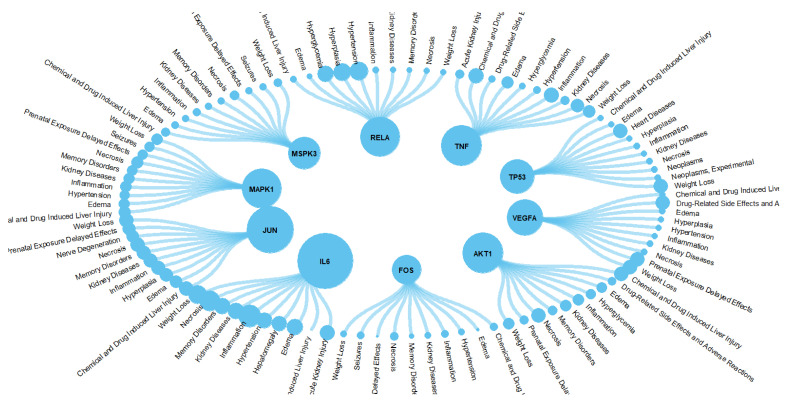
Top 10 hub genes associated with the gene-related diseases.

**Figure 6 ijerph-19-16475-f006:**
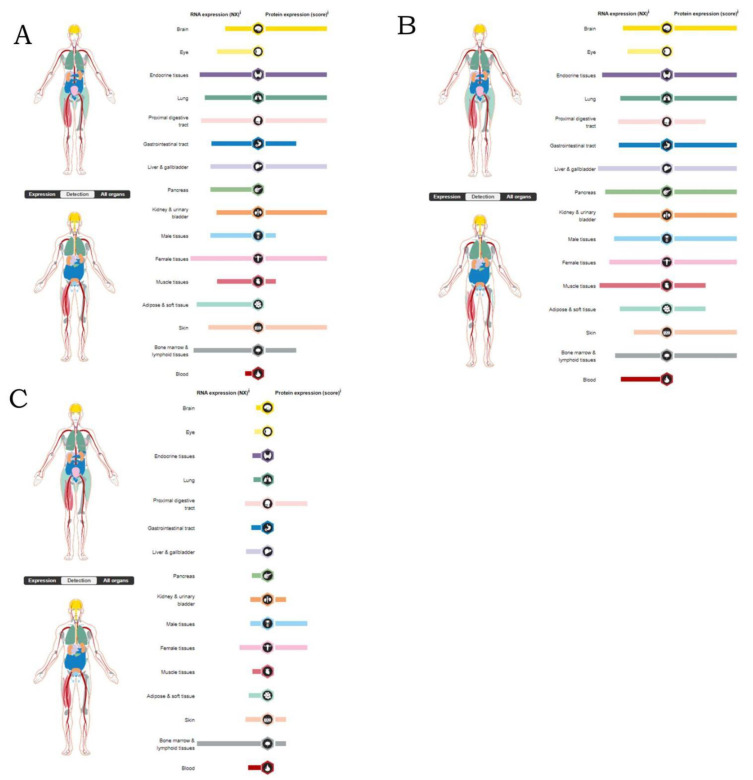
The distribution of genes that are related to colonic neoplasms and breast neoplasms in human organs: (**A**) JUN; (**B**) AKT1; (**C**) TP53.

## Data Availability

Data and materials are available for manuscript.

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
