# Peer review of "Comprehensive Analysis of the Association between Human Diseases and Water Pollutants"

_ijerph, 2022, doi:10.3390/ijerph192416475_

Round 1
Reviewer 1 Report
General Comments
The authors provide an analysis of the influence of water pollutants on human health through study of gene interactions. Overall, the manuscript is difficult to interpret due to the extensive grammatical errors that exist throughout. Similarly, the authors fail to provide adequate context for how the analyses were done or the relevance of this study in a broader context. It is well known that water pollutants can cause adverse effects on ecosystem and human health. Therefore, it is not clear how the study presented here advances knowledge within the field. Similarly, many of analyses that are presented within the manuscript are not adequately described, such that it is not clear to the reader how the study was conducted. Therefore, it is difficult to interpret whether the analyses were appropriate to answer the experimental question. The figures that are presented are also of poor quality and visually difficult to inspect. Considering the lack of novelty of the study as presented in the manuscript, it is suggested that the manuscript be rejected from publication. Some specific comments are below.
Specific Comments
Line 24: The final sentence of the abstract refers to the potential effects of plastics in human health, but plastics are not previously mentioned in the abstract. A list of water pollutants is listed in the Methods section of the abstract, but no mention of plastics is included. If the focus of the manuscript is to the on the potential effects of plastics, this should be made clearer in the abstract.
Line 53: Grammatical error. “…affected the human…” should be “…affected humans…”
Line 57: Grammatical error. “…appeared long time ago…” should be “…appeared a long time ago…”
Line 82: Can the authors please provide a list of the 51 chemicals that were studied? This is relevant to understand the relevance of this work in a global context.
Line 83: It is not made clear what “the R” is and how it was used to explore significant interactions with genes in relation to water pollution. More explanation of what R is and how it was used is needed.
Line 84: How are gene interactions being defined in this context? Are these chemicals that are affecting DNA structure, gene expression, mRNA translation? More information is needed.
Line 104: Cytoscope 3.8.2 should be more clearly described.
Line 107: It is not clear what 20 degrees means in this context and why this was considered the threshold for this study.
Figure 2: The diagrams in panels A-C are too small to be read and interpreted by the reader.
Figure 4: The panels D, G, and H cannot be read.
Author Response
Dear editors and reviewers:
Thank you for reviewing our manuscript and providing valuable comments and suggestions. We have carefully considered all your comments in detail, and have revised our manuscript based on comments point-by-point. Again, it is our honor for this work be suggested and judged. If you have any more questions, please do not hesitate to contact us.
Response: Thank you for your valuable comments! In this work, we aim to explore the potential correlation between water pollutants and human diseases. We first obtain the interactive genes of eight types of water pollutants from the CTD database, which includes perfluorinated chemicals, polybrominated diphenyl ethers, phthalate, nanomaterials, insecticides, microcystins, heavy metal and pharmacology. Based on the interactive counts of the interactive genes involved in the water pollutants, we successfully obtain the top interactive genes. Subsequently, we performed the GO and KEGG enrichment analysis. The results revealed that many human diseases are closely associated with the water pollutants, such as colonic neoplasms and breast neoplasms. In addition, the specific water pollutants were also involved in the analysis. Finally, we discovered the genes that play a key role in the water pollutants and colonic neoplasms and breast neoplasms. TP53, RELA and JUN were considered to be closely associated with water pollutants and colonic neoplasms. In addition, TP53 and RELA were found to be the most relative to breast neoplasms. A previous published article entitled “An integrative toxicogenomic analysis of plastic additives” in the journal of “Journal of Hazardous Materials” also focused on the association between plastic additives and human health by bioinformatic analysis. We think it is very innovative to evaluate the correlation between pollutants and human health by bioinformatic analysis.
For the specific comments, we have corrected all the grammatic errors in this manuscript. In addition, a list of the 51 chemicals have also been provided in the supplemental files. Also, the expressions that may have caused ambiguity have been corrected. Based on the previous study, the degree of 20 was considered as the hub genes for the PPI network. (Zhang X, Lu Z, Ren X, Chen X, Zhou X, Zhou X, Zhang T, Liu Y, Wang S, Qin C. Genetic comprehension of organophosphate flame retardants, an emerging threat to prostate cancer. Ecotoxicol Environ Saf. 2021 Oct 15;223:112589. doi: 10.1016/j.ecoenv.2021.112589. Epub 2021 Aug 3. PMID: 34358932.). For figure 4 and figure 2, we have enlarged the figure to make it clearer.
Reviewer 2 Report
Interesting article. The manuscript should revise for standard English, and abstract, Introduction, result and conclusions part should improve.
Background, Methods, results and conclusion word no need mention in the abstract part.
Line No. 33: by the World Health Organization
Line No. 35: about 3.4 million 34 people die each year
Line No. 43: pollutants were wide varieties
Line No 61: closely associated with many diseases
Line no. 64: literature based
Line No. 65: accumulated through the continuous aquatic food chain
Line No. 68: understanding about how environmental exposures
Line No. 105: defined as genes highly interconnected with nodes in a module
Line No. 111: Venn diagrams were used
Line No. 169: Study suggested
Line No. 207: common water pollutants were classified
Line No. 210: venn diagrams were conducted
Line No. 262: important resources
Line No. 269: under threat
Line No. 270: water resources
Line No. 314: phthalate may demonstrate
Author Response
Dear editors and reviewers:
Thank you for reviewing our manuscript and providing valuable comments and suggestions. We have carefully considered all your comments in detail, and have revised our manuscript based on comments point-by-point. Again, it is our honor for this work be suggested and judged. If you have any more questions, please do not hesitate to contact us.
Response: Thank you for reviewing our manuscript and providing valuable comments and suggestions. We have carefully considered all your comments in detail, and have revised our manuscript based on comments point-by-point. Again, it is our honor for this work to be suggested and judged. If you have any more questions, please do not hesitate to contact us.
We are very sorry for so many grammatic errors. In the revised manuscript, we have corrected the grammatical errors. The grammatical errors mentioned above have been corrected. In addition, the abstract part has also been improved.
Reviewer 3 Report
The work is very interesting, well-designed, and well-presented. I would suggest the work get accepted following a minor revision, addressing my comments:
1. It is almost impossible to read the descriptions or numbers in all figures. Please use larger fonts to make them readable.
2. Figure 6 is so small that I cannot see the details of the bodies, please enlarge them.
3. Descriptions for Figure 3 should be expanded, as they are very brief.
4. Please discuss the utility of emerging technologies such as microfluidics for the detailed analysis of pollutants in water.
Author Response
Dear editors and reviewers:
Thank you for reviewing our manuscript and providing valuable comments and suggestions. We have carefully considered all your comments in detail, and have revised our manuscript based on comments point-by-point. Again, it is our honor for this work be suggested and judged. If you have any more questions, please do not hesitate to contact us.
Response: Thank you for reviewing our manuscript and providing valuable comments and suggestions. We have carefully considered all your comments in detail, and have revised our manuscript based on comments point-by-point. Again, it is our honor for this work to be suggested and judged. If you have any more questions, please do not hesitate to contact us.
In this manuscript, we uploaded the figures through the system, the resolution may reduce after generating the PDF format. In the revised manuscript, we have added the figures in the manuscript. In addition, the figures have also been enlarged to make them clearer. In figure 3, we mainly aim to find key genes for the further exploration of the roles of interactive genes in water pollutants. The PPI network was applied to evaluate whether the proteins encoded by water pollutants-related genes have significant interactions. In the results part, we have provided more descriptions of figure 3.
Round 2
Reviewer 1 Report
The manuscript has been significantly revised to be considered for publication.
Reviewer 2 Report
Please check Plagiarism.